



# Development and application of a mass closure PM$_{2.5}$ composition online monitoring system

Cui-Ping Su[1], Xing Peng[1], Xiao-Feng Huang[1*], Li-Wu Zeng[1], Li-Ming Cao[1], Meng-Xue Tang[1], Yao Chen[1], Bo Zhu[1], Yishi Wang[2], and Ling-Yan He[1]

[1]Key Laboratory for Urban Habitat Environmental Science and Technology, School of Environment and Energy, Peking University Shenzhen Graduate School, Shenzhen, 518055, China
[2]Princeton Day School,650 Great Road, Princeton, NJ 08540

*Correspondence to*: Xiao-Feng Huang (huangxf@pku.edu.cn)

**Abstract.** Online instruments have been widely applied for the measurement of PM$_{2.5}$ and its chemical components; however, these instruments have a major shortcoming in terms of the lack or limited number of species in field measurements. To this end, a new mass closure PM$_{2.5}$ online integrated system was developed and applied in this work to achieve more comprehensive information on chemical species in PM$_{2.5}$. For the new system, one isokinetic sampling system for PM$_{2.5}$ was coupled with an

aerosol chemical speciation monitor (Aerodyne, ACSM), an Aethalometer (Magee, AE-31), an automated multimetals monitor (Cooper Corporation, Xact-625), and a hybrid synchronized ambient particulate real-time analyzer monitor (Thermo Scientific, SHARP-5030i) to enable high-resolution temporal (1 h) measurements of organic matter, SO$_4^{2-}$, NO$_3^-$, Cl$^-$, NH$_4^+$, black carbon and important elements as well as PM$_{2.5}$ mass concentrations. The new online integrated system was first deployed in Shenzhen, China to measure the PM$_{2.5}$ composition from 25 September to 30 October 2019. Our results showed that the average PM$_{2.5}$

concentration in this work was 33 μg m$^{-3}$, and the measured species well-reconstructed the PM$_{2.5}$ and almost formed a mass closure (94%). The multilinear engine (ME-2) model was employed for the comprehensive online PM$_{2.5}$ chemical dataset to apportion the sources with predetermined constraints, in which the organic ion fragment m/z 44 in ACSM data was used as the tracer for secondary organic aerosol (SOA). Nine sources were determined and obtained reasonable time series and diurnal variations in this study, including identified SOA (23%), secondary sulfate (22%), vehicle emissions (18%), biomass burning

(11%), coal burning (8.0%), secondary nitrate (5.3%), fugitive dust (3.8%), ship emissions (3.7%), and industrial emissions (2.1%). The potential source contribution function (PSCF) analysis indicated that the major source areas could be the region north of the sampling site. To the best of our knowledge, this is the first system that can perform online measurements of PM$_{2.5}$ components with a mass closure, thus providing a new powerful tool for PM$_{2.5}$ long-term daily measurement and source apportionment.



## 1 Introduction


PM$_{2.5}$ (aerodynamic particles smaller than 2.5 microns in size) is currently a serious problem because its chemical components have strong harmful effects on visibility (Campbell et al., 2018), radiative forcing (Gao et al., 2017), human health (Puthussery, et al., 2018; Zhang et al., 2019), climate change (Glotfelty et al, 2014), and ecosystems (Diemoz et al., 2019; Amarloei et al., 2020). PM$_{2.5}$ contains many types of components, such as trace elements, water-soluble inorganic ions, elemental carbon (EC),

black carbon (BC), organic carbon (OC) or organic matter (OM), etc. (Budisulistiorini et al., 2014; Jayarathne et al., 2018). Accurately measuring PM$_{2.5}$ and chemical species are essential for identifying their sources, assessing their negative effects (e.g., radiative forcing, human health, etc.), and developing effective abatement strategies (Huang et al., 2018; Hand et al., 2019).

Various online measurement techniques for PM$_{2.5}$ and chemical species have been developed and widely applied to get their

detailed chemical and physical information in recent years because of their high time resolution (1 h or less) (Vodicka et al., 2013; Li et al., 2017). For example, the Aerodyne Aerosol Chemical Speciation Monitor (ACSM) has been used to determine OM, inorganic aerosol constituents in PM$_1$ or PM$_{2.5}$ (Budisulistiorini et al., 2014; Zhang et al., 2017). Water-soluble ions in PM$_{2.5}$ can be determined by several online instruments using ion chromatography (IC) methods, such as Particle Into Liquid Sampler-ion chromatography (PILS-IC), the Monitor for AeRosols and GAses (MARGA), the in situ Gas and Aerosol

Composition (IGAC), and ambient ion monitor (AIM, URG Corporation, USA) (Kuokka et al., 2007; Harry et al., 2007; Ellis et al., 2011; Yu et al., 2018). A semicontinuous OC/EC analyzer was developed by Sunset Laboratory for the measurement of OC and EC (Vodicka et al., 2013). Trace and crust elements can be measured by an automated multimetals monitor (Xact-625) using the X-ray fluorescence method (XRF) (Yu et al., 2019). However, a major shortcoming in these instruments is the lack or limited species from field measurements. To obtain more comprehensive information on chemical species in PM$_{2.5}$, some

previous studies have employed several online instruments to simultaneously measure the organic carbon, element carbon, important elements, and water-soluble inorganic ions in PM$_{2.5}$ (Gao et al., 2016; Peng et al., 2016b; Wang et al., 2018; Huang et al., 2019; Liu et al., 2019). Although major species of PM$_{2.5}$ have been measured in these research studies, the sum concentration of all the determined species was not close to the PM$_{2.5}$ mass concentrations, which was partly because OM has not been measured, which is required to measure PM$_{2.5}$ mass closure (Hand et al., 2011; Wang et al., 2018). OM can be

calculated by multiplying the OC concentrations by a multiplier that is an estimation of the average molecular weight per carbon weight for the organic aerosol. This multiplier has been investigated in previous studies (Turpin and Lim, 2001; Boris et al., 2019; Chow et al., 2019) and is widely used to estimate OM concentrations (Chow et al., 2015). This estimation method has large uncertainty because the multiplier varies with time and location and ranges from 1.2 to 2.6 (Chow et al., 2015). Therefore, accurately characterizing the OM and other major species of PM$_{2.5}$ is important for computing the PM$_{2.5}$ mass

closure and estimating their contribution to PM$_{2.5}$ mass concentrations. To this end, we developed a new mass closure PM$_{2.5}$ online integrated system, where one isokinetic sampling system for PM$_{2.5}$ is coupled with the ACSM, AE-31, Xact-625, and SHARP-5030i to enable high time resolution (1 h) measurements of organic matter, inorganic ions, black carbon, important

elements, and PM$_{2.5}$ mass concentration. Compared with other online instruments (e.g., AMS, semicontinuous OC/EC analyzer, ACSM, etc.), the new online integrated system could fully measure the chemical species and better achieve PM$_{2.5}$ mass closure.

The SHARP-5030i monitor uses particle light scattering and Beta Attenuation Monitors, which are recommended for use in PM$_{2.5}$ monitoring networks in China. Therefore, obtaining the comprehensive chemical species of PM$_{2.5}$ measured by the SHARP-5030i monitor and apportioning the PM$_{2.5}$ sources based on the mass closure data could be directly used to control PM$_{2.5}$. Researches have indicated that SOA is an important contributor of PM$_{2.5}$ and that it originated from the oxidation of gas phase precursors (Carlton et al., 2009; Huang et al., 2018). It is currently difficult to measure SOA, and most estimates are

indirect and use the OC/EC ratio, receptor models (e.g., CMB and PMF), and chemical transport models (Murphy and Pandis, 2009; Cesari et al., 2016; Al-Naiema et al., 2018). The ACSM can also measure the nucleus-mass ratios of fragmented ions, such as m/z44 ($CO_2^+$), which is a good indicator for SOA (Zhu et al., 2018). We propose an improved source apportionment method using m/z44 as the prior information to introduce into the receptor model to solve the collinear problem and estimate SOA contributions to PM$_{2.5}$ mass concentrations.

Here, the new online integrated system was deployed in Shenzhen, China in autumn for approximately one month, and its field performance was evaluated by comparing the results with other measurements of simultaneously collected data. The online receptor dataset also used to fully resolve the PM$_{2.5}$ sources using receptor model. We used mass spectra information to constrain the SOA and successfully estimated SOA contributions to PM$_{2.5}$.

## 2 System design and integration

### 2.1 Objectives


Our study aimed to integrate the most reliable and stable online instruments under the current technology development level for components of PM$_{2.5}$. The same Cut Cyclone values could ensure the comparability of different instruments, and the sampling manifold was developed to ensure that particles enter each online monitoring instrument with minimum loss. The flowrate, pipeline turbulence and dehumidification effect need to be fully considered during the research and development of

the manifold. The data analytics platform was designed to integrate and manage data automatically.

Based on the objectives above, the integrated system included three parts: sampling module, instrumentation, and data analytics platform, which was used to store data and perform analyses. The sampling module included a size-selective sampling head and a tube to deliver aerosol to the instrument. The instrumentation part included the online monitoring equipment for PM$_{2.5}$ and its chemical components.

### 2.2 Instrument selection


To realize high temporal-resolution, precise, stabilized, and mass-closure chemical measurements of PM$_{2.5}$ concentrations, four typical online instruments were selected to integrate into the new system: ACSM, Xact-625, AE-31, and SHARP-5030i.



The ACSM was selected for monitoring the mass concentrations of ambient organic aerosol and inorganic ions ($SO_4^{2-}$, $NO_3^-$, $NH_4^+$, and $Cl^-$) because of its small volume, low cost, operational stability and high resolution (18 min) (Nga et al., 2011).

Xact-625 was selected to obtained the hourly element concentration in $PM_{2.5}$. The quantification of 21 elements in $PM_{2.5}$, such as As, K, Co, Ti, Ca, Cr, Fe, Hg, V, Mn, Sn, Ni, Cu, Sb, Zn, Se, Mo, Cd, Ba, and Pb, was performed using nondestructive energy-dispersive X-ray fluorescence, which was certified by the United States Environmental Protection Agency IO3.3(Ji et al., 2018). Modifications were conducted in the Xact-625 design to realize the measurement of Si, because it is a tracer of fugitive dust source (Taylor and Mclennan,1995). The BC mass concentration represented a large fraction in the atmosphere (Prasad et al., 2018), and the advanced AE-31 with 5 min temporal resolution was used for continuous monitoring of the BC mass concentration at seven wavelengths viz., 370, 470, 525, 590, 660, 880, and 950 nm. Due to strong absorption of BC aerosols at 880 nm, the mass concentrations measured at this wavelength were considered as a standard for BC measurements (Prasad et al., 2018). To monitor $PM_{2.5}$, a SHARP-5030i instrument was used. SHARP-5030i measures $PM_{2.5}$ mass concentrations based on the principles of particle light scattering (nephelometer) and beta attenuation. The monitor measures $PM_{2.5}$ mass concentrations every minute and reports hourly averages (Su et al., 2018). When the time resolution is 1 hour, the minimum detection limit is less than 0.5 μg m$^{-3}$. (Ji et al., 2018).

## 2.3 Design of the rack and sampling module

The sampling module mainly involved the selection of the $PM_{2.5}$ Very Sharp Cut Cyclone (VSCC) particle size separator and the design of the particle isokinetic sampling manifold. A cyclone (sharp cut cyclone 2000-30EC, URG Inc. US) with a flow rate of 42 L per minute was selected, which was based on the total flow of the sampling instruments above. The main technical challenge in designing the sampling module was the sampling manifold, which needed to reduce turbulence due to the different flow rate of instruments. To solve this problem, a particle isokinetic sampling manifold was designed, which was equivalent to thin-walled, sharp-edged, isokinetic inlets (the velocity ratio of the velocity outside of the isokinetic inlet and the average sampling velocity in the inlet is 1) that were sampled isoaxially from vertical airflows under laminar flow conditions (Reynolds number<2000), which is proven to sample particles with a transmission efficiency of close to 100% (Okazaki et al., 1987). As shown in Fig. 1a, the main pipe diameter was calculated according to the total flow rate of the integrated instruments and designed to be 31 mm to ensure that airflows under laminar flow conditions when it reaches the inlet of the isokinetic sampling tube. The gradual expansion angle of the gradual expansion channel in the pipeline was 7° to prevent the gas from separating from the pipe wall, and temperature, humidity and flow sensors were equipped on sample tubes to monitor the gas state. The sampling flow of the sample tube was large to meet the needs of integration. After the test, the relative to standard deviation of the mass concentration among each instrument was within 5%. To integrate the instruments, a rank was designed according to the size of integrated instruments, on which aluminum plates were laid to form a platform for placing the instruments (Fig. 1b). A physical picture of the system is shown in Fig. S1.





(a)  (b)

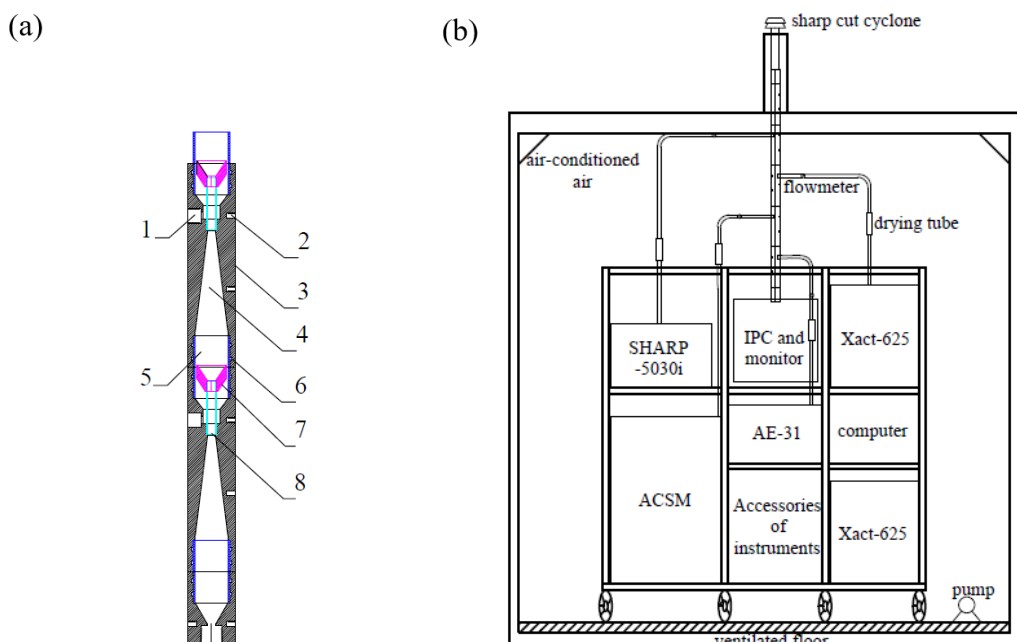

**Figure 1. Structure diagram of the particle isokinetic sample tube (1-Aerosol outlet; 2-Screws for fixing; 3-Main pipe; 4-Gradual expansion channel; 5-Connecting pipe; 6-O-ring; 7-Isokinetic sampling tube; and 8-Connecting pipe) (a). Sketch map of integrated system (IPC=Industrial Personal Computer) (b).**

## 2.4 Design of the data analytics platform

To integrate online data, a data analytics platform (Fig. S2) was designed and used to export data from the integrated instruments with a uniform time resolution of 1 hour, and it was also able to display the variation trends of $PM_{2.5}$ and its chemical components in real time and back up these data in an SQL server database.

## 3 Application of the online integrated system

### 3.1 Sampling site

The sampling site was in the Peking University Shenzhen Graduate School (22°35' 45" N, 113°58'23" E) which is in the western urban area of Shenzhen, Shenzhen is an international metropolis in China, surrounded by the south China sea to the southeast and industrial cities such as Dongguan and Huizhou to the west and north. It is located in the subtropical marine climate zone, with more rainfall in summer influenced by monsoons and heavy pollution in autumn and winter due to the influence of inland air mass. The sampling site was surrounded by a large reservoir and recuperation base, with a high vegetation coverage ratio. There is no obvious pollution source except a city road about 100m away. The measurement campaign took place from 25 September to 30 November 2019, when the pollution usually started to worsen in Shenzhen. The meteorological conditions during the campaign were shown in Fig. S3. The average ambient temperature and relative humidity





were 26.7±3.8°C and 71.9±19.0%, respectively. The wind speed ranged from 0.1 to 6.9 m s$^{-1}$, with an average value of 0.6±0.5 m s$^{-1}$. The sampling site was mainly influenced by wind from the northeast and northwest.

### 3.2 Sampling instruments

During the sampling period, in addition to the online integrated system, another AE-31 and MARGA system were set for independent measurements at the same site. In additional, the offline PM$_{2.5}$ were simultaneously conducted every two days and each sample was collected for 24 hours. The sampling instruments and analytical methods of PM$_{2.5}$ chemical composition refer to previous studies by Huang et al. (2018). The performance of the online integrated system was thoroughly evaluated by comparison with the measurements above.

### 150   3.3 Source apportionment modeling

#### 3.3.1 Multilinear engine (ME-2) model

ME-2 is a factorization tool for multivariate factors, which was developed by Paatero in 1999(Paatero, 1999). The basic principle of ME-2 can be expressed that the sample data matrix X are grouped into two constant matrix G and F, as follow: and make sure that all elements of G and F are non-negative.

$X = G \times F + E$                                          (1)

G and F are factor contribution and factor profile, respectively, which are non-negative. E is residual error matrix that the difference between measured value and the model value (Paatero and Tapper, 1994).

The objective of the ME-2 model is to minimize the equation Q, as follow:

$$Q = \sum_{i=1}^{m} \sum_{j=1}^{n} (e_{ij}/u_{ij})^2 \tag{2}$$

where $e_{ij}$ is the element of the matrix E and $u_{ij}$ is the error/uncertainty of the measured concentration $x_{ij}$. Source Finder (SoFi) software was developed by the Paul Schell Institute (PSI) in Switzerland, which can provide some constrained factors to realize the source constraint of ME-2(Canonaco et al., 2013). The species uncertainties ($u_{ij}$) are estimated as follows:

$$u_{ij} = k_j \times x_{ij} \tag{3}$$

where the $x_{ij}$ is the concentration of the species and $k_j$ is the $j^{th}$ species' error fraction (Huang et al., 2018), which is usually

thought to be between 5% and 30% (Zou, 2018). Through testing, the $k_j$ in this study was set as 20%, and some species were expanded or decreased according to the results of model operation. When the species concentration is greater than or equal to the detection limit (DL), the uncertainty can be estimated according to the Eq. (3). Conversely, the concentration was treated by 1/2 DL and its uncertainty was set as 5/6 DL. Missing data were treated by the species' average value (geometric), and their uncertainties were assumed to be 4 times of the average value.



### 3.3.2 Constraint setup in ME-2 modeling

SoFi was performed to initiate and control ME-2 model, and it was run for the online datasets observed by integrated system in this study. Species that satisfied one of the following conditions were excluded: (1) more than 40% data were smaller than the DL of species, (2) less relevant to other species($r^2 \leq 0.4$), and (3) low concentration and can not to be used as a tracer for pollution sources. Therefore, 13 species, accounting for 98.6% total mass concentrations of the measured species(except m/z 44), were input into the models: OM, BC, $Cl^-$, $NO_3^-$, $SO_4^{2-}$, $NH_4^+$, K, Ca, Fe, Zn, V, m/z44.

Compared with some other interrelated studies, fragments of organic matter with m/z44 were input into the model to track the SOA. m/z44 is mainly composed of fragments of carboxylic acid ions $CO_2^+$, which is produced from the cracking of carboxylic acid in organic aerosols and can represent the strength of oxidation of organic aerosol (Hu, 2012). Therefore, m/z44 was used to characterize the SOA with strong oxidization.

Based on the experience of our previous studies on source resolution in Shenzhen (Huang et al., 2018; Shen et al., 2019) and the introduction of SOA tracers, four factors were constrained as listed in Table 1: SOA, secondary sulfate, secondary nitrate, and fugitive dust. For the secondary sulfate and nitrate sources, the primary source components, OM, and m/z44 in the factors were set to 0 while $NH_4^+$ and $SO_4^{2-}$/$NO_3^-$ were not limited. For the SOA, OM and m/z44 were not limited. Based on a study of fugitive dust source profiles (Peng et al., 2016a), the earth crust elements (e.g., Si, Ca, and Fe) and OM were not limited in the fugitive dust.

**Table 1. Detailed constraints of the four factors using in ME-2.**

| Factors | OM | BC | $Cl^-$ | $SO_4^{2-}$ | $NO_3^-$ | $NH_4^+$ | K | Si | Ca | Fe | Zn | V | m/z44 |
|---|---|---|---|---|---|---|---|---|---|---|---|---|---|
| Secondary sulfate | 0 | 0 | 0 | - | 0 | - | 0 | 0 | 0 | 0 | 0 | 0 | 0 |
| Secondary nitrate | 0 | 0 | 0 | 0 | - | - | 0 | 0 | 0 | 0 | 0 | 0 | 0 |
| SOA | - | 0 | 0 | 0 | 0 | 0 | 0 | 0 | 0 | 0 | 0 | 0 | - |
| Fugitive dust | - | 0 | 0 | 0 | 0 | 0 | 0 | - | - | - | 0 | 0 | 0 |

### 3.3.3 Potential source contribution function (PSCF) analysis

The PSCF was based on the analysis results of a backward trajectory, and it was used to analyze the potential regions of the possible sources of $PM_{2.5}$. Unlike the backward trajectory, which could only obtain the pathways of the air masses, PSCF can also analyze the regional transmission contribution of pollutants from different sources in the region of a given day. In TrajStat software, the movement of an air mass is described by segment endpoints of latitude and longitude coordinates (Wang et al., 2009). Before performing the PSCF analysis, the coverage area of the trajectory was 15-40°N and 90-140°E and the grid resolution was set to 0.4°×0.4° grid cells. The PSCF values for grid cells were calculated by the ratio of the number of





contamination trajectory full in the grid cells and the number of all trajectories passing through the area. The PSCF can be

defined as follows:

$$PSCF_{ij} = \frac{m_{ij}}{n_{ij}} \tag{4}$$

where: i and j are latitude and longitude, respectively; $PSCF_{ij}$ is the PSCF value for the $ij^{th}$ grid cell; $m_{ij}$ represents the number of endpoints corresponding to specific source contribution above the threshold criterion, which is set to the $75^{th}$ percentile of each source contribution; and $n_{ij}$ represents the total number of endpoints falling in the grid cell ij.

To avoid the uncertainty caused by the limited number of endpoints in a single grid, a weight function $W_{ij}$ is also considered here as shown in Eq. (5) and Eq. (6). For example, when the total number of trajectory endpoints in each grid ($n_{ij}$) is higher than 20 (the average endpoint) but less than 40 (4 times of average endpoint), the weighted PSCF value (WPSCF) of that grid is reduced by multiplying its original value by 0.7. The WPSCF results were used in the following discussion and called PSCF in this work.

$$WPSCF = PSCF_{ij} \times W_{ij} \tag{5}$$

$$W_{ij} = \begin{cases} 1.00 & n_{ij} > 40 \\ 0.70 & 20 < n_{ij} \le 40 \\ 0.42 & 10 < n_{ij} \le 20 \\ 0.05 & n_{ij} \le 10 \end{cases} \tag{6}$$

## 4 Results and discussion

### 4.1 Overview of chemical species and PM$_{2.5}$ concentrations

The time series of hourly PM$_{2.5}$ concentration and chemical species and the average concentration results are shown as Fig. 2a

and Table S1, respectively. Based on hourly data, PM$_{2.5}$ concentrations ranged from 5 to 107 µg m$^{-3}$, with an average value 33±14 µg m$^{-3}$, which was lower than the NAAQS (National Ambient Air Quality Standards, China, 2012) for annual PM$_{2.5}$(35 µg m$^{-3}$). OM (14.1±7.4 µg m$^{-3}$), SO$_4^{2-}$ (8.6±3.3 µg m$^{-3}$), NO$_3^-$ (1.8±1.9 µg m$^{-3}$), NH$_4^+$ (3.8±1.7 µg m$^{-3}$), and BC (2.1±1.0 µg m$^{-3}$) were the major species, with average percentages of 41%, 25%, 5.3%, 11%, and 6.4% in PM$_{2.5}$, respectively. The major elements included Si (1.3%), K (1.3%), Fe (0.91%), Ca (0.35%), Zn (0.31%), and the trace elements accounted for 0.84% in

PM$_{2.5}$ (Fig. 3).

### 4.2 Evaluation of PM$_{2.5}$ mass closure

To access the accuracies and mass closure of the receptor data from the integrated system, we reconstructed the PM$_{2.5}$ mass concentration according to the summation of chemical components. The correlation between the reconstructed and measured PM$_{2.5}$ mass concentration was shown as Fig. 2b. Reliable data were measured by the online integrated system and supported

by the good correlation (correlation coefficient squared r$^2$=0.89) and the slope (0.94), which was close to 1, and the results indicated that the integrated system can measure 94% of the particulate matter components and achieve mass closure. The





online integrated system can measure more chemical components than other online instruments, for example, online continuous monitoring and analysis systems (AIM-URG9000d, USA) for water-soluble components of fine particulate, which can measure water-soluble ion contents and accounted for about 40% of $PM_{2.5}$ mass concentrations (Yuan et al., 2018). A

semicontinuous OC/EC analyzer with 1 h time resolution could obtain the concentrations EC and OC, which accounted for approximately 30% of the $PM_{2.5}$ (Liu et al., 2017). Wang et al. (2018) measured the chemical compositions using MARGA, a semicontinuous OC/EC analyzer, and Xact-625 and reported the sum mass concentration of measured species accounted for ~78% of $PM_{2.5}$.

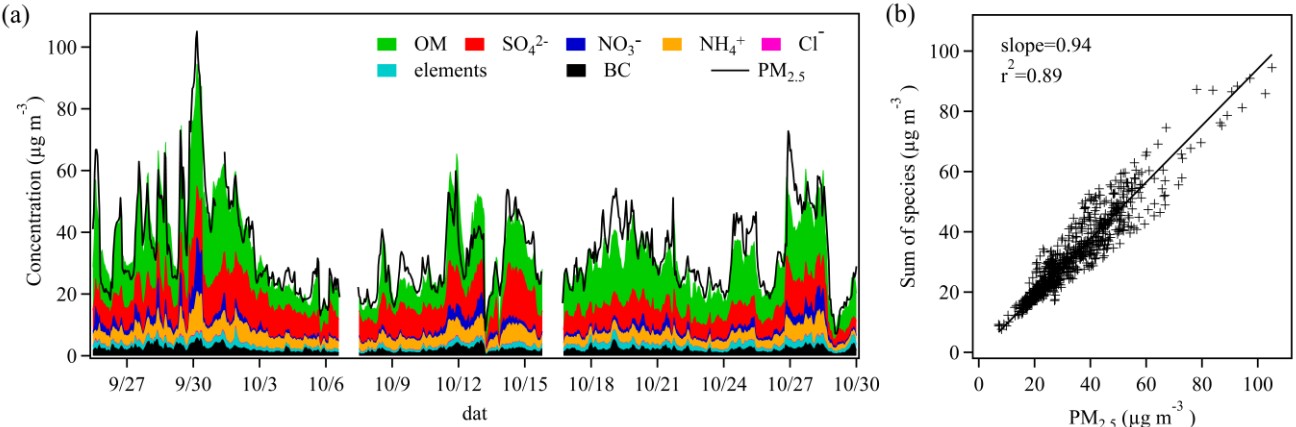

**Figure 2. The trend in hourly $PM_{2.5}$ and chemical species during the sampling camaign (a). The correlation between $PM_{2.5}$ reconstructed and measured (b). Note that the term of "Sum of species" means the total mass concentration of chemical species from online integrated system, and "$PM_{2.5}$" indicates mass concentration of SHARP-5030i $PM_{2.5}$.**

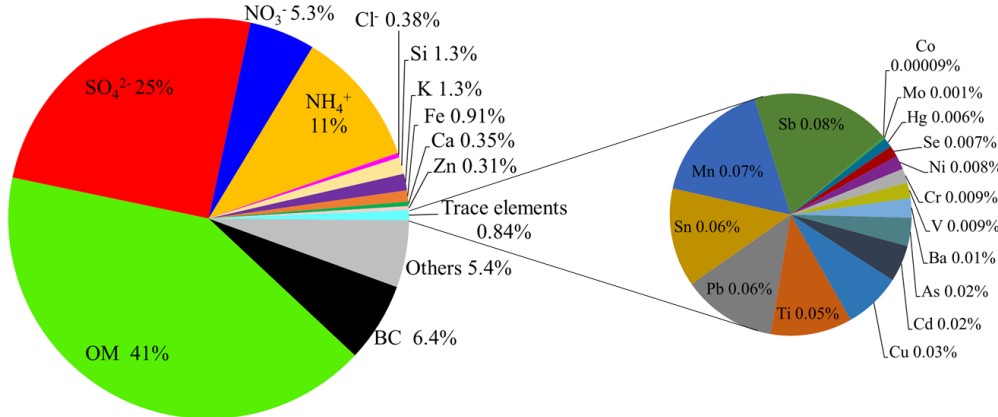

**Figure 3. Chemical compositions of average $PM_{2.5}$ during the sampling campaign.**



### 4.3 Comparison with colocated measurements

The average of PM$_{2.5}$ concentrations from offline sampling was 32 µg m$^{-3}$, which was consistent with the integrated system, and the chemical components in PM$_{2.5}$ also obtained similar results (Fig. 4), in which OM measured by online integrated and offline system were 14.6 µg m$^{-3}$ and 14.2 µg m$^{-3}$, while SO$_4^{2-}$, NO$_3^-$, NH$_4^+$ were 7.9%, 19.7%, and 5.0% higher in online integrated system, respectively, which is possibly due to the volatilization and loss of sample during the process of offline sampling, storage, and analysis. The BC measured by the online integrated system was 8.4% higher than the EC in the offline system, and the differences of important elements were relatively small. In general, the online and offline measurement were close to each other, which indicated that the online integrated system was accurate to some extent.

Comparing the data with the results of other method analysis usually be used to further examine the reliability of online instruments. The BC showed high correlation with measurements of another AE-31 (r$^2$=0.99). On average, the mass concentrations of BC measured by integrated system reports 95% of those measured by the independent instruments (Fig. S4), suggesting the sampling model of online integrated system could avoid sampling loss effectively and was reliable. This slight underestimation of BC mass might be primarily due to the distance between sampling ports or the systematic errors between instruments.

Figure S4 shows the intercomparisons of the measurements by the ACSM with MARGA, including SO$_4^{2-}$, NO$_3^-$, NH$_4^+$, and Cl$^-$. Overall, the three ions measured by the ACSM correlated well with MARGA (r$^2$=0.94-0.96). The sulfate average concentrations between ACSM and MARGA were consistent (slope=0.97). The ammonium concentrations measured by ACSM accounted for 88% of that determined by MARGA, which might relate to the aerodynamic lens transmission efficiency of ACSM. When the particle size was less than 300 nm, the transmission efficiency was lower (Middlebrook et al., 2012). For nitrate, the ACSM measurements approximately 64% of what was reported by the MARGA. ACSM measured a lower Cl$^-$ concentration (slope=0.39) and a relatively poor correlation with MARGA's Cl$^-$ (r$^2$=0.42). In this study, the ACSM evaporator was ~550℃, and some components such as NaNO$_3$, NaCl, and Mg(NO$_3$)$_2$ in the aged sea salt and mineral dust may not be detected due to the lack of gasification at this temperature. The low concentration of nitrate and chloride ions measured by the ACSM may be caused by the presence of sea salt or crustal particles in the collected samples. Another study also found a large difference between ACSM and MARGA when measuring chloride and nitrate (Zhang et al., 2017).

K obtained good correlation between the Xact-625 and MARGA (r$^2$=0.75, Fig. S4), and the average ratio of Xact-625 to MARGA was 1.18, which might be caused by the slight difference between the K element and K$^+$ in PM$_{2.5}$ as well as the differences of measurement principles. This result suggested the reliability of the Xact-625 in measuring elements.





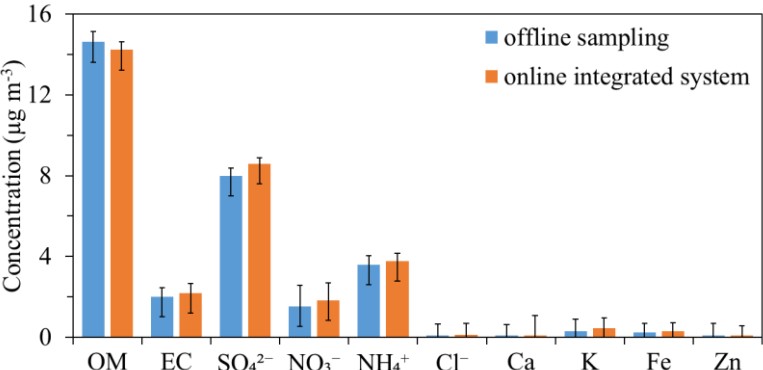

**Figure 4. Concentrations of major chemical compositions of PM$_{2.5}$ measured by online Integrated system and offline sampling. Note that the EC in the online integrated system was referred to as BC.**

### 4.4 Source apportionment analysis

#### 4.4.1 Source apportionment results

In this study, the SoFi tool was used to run the ME-2 model based on the predetermined constraints (Table 1) and to realize the source apportionment, 7-10 factors solution were tried, and the result of nine factors showed the best source profiles ($Q_{true}/Q_{exp}$ ratio was 1.2), with the scaled residuals distributed between -3 and +3 approximately symmetrically. when the factor number was less than 9, the $Q_{ture}/Q_{exp}$ was too high, while the factor number was more than 9, a meaningless high OM source would appear. (Fig. S5). The total mass concentration of the components (m/z44 was not included here because it was the part of OM) input to the ME-2 model was significantly consistent with the total mass concentration of the model reconstruction ($r^2$=0.99, slope=0.98).

Profiles of each factor were shown Fig. 5a. From top to bottom, factor 1 had large percentage explained variation (EV) values of SO$_4^{2-}$ and NH$_4^+$, and was identified as secondary sulfate (Huang et al., 2018). Factor 2 was related to secondary nitrate, due to significant EV values of NO$_3^-$ and NH$_4^+$ (Huang et al., 2018). Factor 3 was identified as SOA, which has a significant EV value of m/z44 and OM, m/z44 had high concentration in this factor, thus suggesting it is a good tracer of SOA. Previous study tried to identify SOA using receptor model and indicated SOA mixed with secondary sulphate and/or secondary nitrates due to the lack of tracer (Huang el al. 2018), in this study, the effective tracer of SOA was introduced into the model resulting in the SOA was identified separately. Factor 4 was associated with coal burning due to the high EV value of Cl$^-$, which mainly come from coal in Shenzhen, besides OM, BC, and SO$_4^{2-}$ were typical combustion products (Li et al., 2016). Factor 5 was explained as fugitive dust due to its high EV values of Si and Ca, which came from the earth crust and building dust, respectively (Feng et al., 2003). The integrated system does not involve the measurement of oxygen (O) and Al. Instead, the contents of O and Al in factor 5 were estimated indirectly based on the abundance of the three (Al, O, Si) in the crust. (Taylor and Mclennan, 1995). Factor 6, with high loading of K, was biomass burning (Yamasoe et al., 2000). Factor 7 was identified as vehicle emissions due to the high concentrations of OM and BC related to oil combustion. This factor also showed certain





loading of Fe might come from  tires and the brake wear of vehicle (He et al., 2011; Yuan et al., 2006). Factor 8 displayed large EV values of Zn, and Fe and Ca also had high enrichment factors. The Zn might have been due to industrial sources,

such as plastic incineration, coating and metallurgy (Zabalza et al., 2006; Cruz Minguillón et al., 2007); the Ca might come from the soot produced by the inferior coal and diesel combustion in the industry (Lewis and Macias, 1980); and the Fe might be related to ceramic production (Querol et al., 2007) and thus was considered to be related to industrial processes. Factor 9 was ship emissions due to a large EV values of V. V is mainly produced by burning of heavy oil, which is the fuel of ships in the port of Shenzhen (Huang et al., 2014).

High time resolution of source contributions could be output when source apportionment was performed using online datasets. Fig. 5b shows the time series of each source as well as their corresponding tracers during source resolution in Shenzhen. The sources had a very good correlation with the tracer species ($r^2$=0.79-0.99), suggesting that the model could identify representative pollution factors. Compared with primary sources, secondary sources fluctuate more dramatically with time. Among all the sources, the trend of secondary sulfate was closer to $PM_{2.5}$, reflecting that it was the one of the main contributors

to $PM_{2.5}$.

Figure 6a showed the average contributions of the $PM_{2.5}$ sources during the sampling period. It can be seen that the SOA, secondary sulfate, vehicle emissions, and biomass burning were the main sources in Shenzhen, contributed 23%, 22%, 18%, and 11% of $PM_{2.5}$ mass concentration, respectively. The total contribution above four sources was more than 70%. The contributions of the other source were smaller (2%-8%), from large to small were coal burning (8%), secondary nitrate (5.3%),

fugitive dust (3.8%), and ship emissions (3.7%). The unidentified sources accounted for 3.4%, which was caused by a combination of residuals from ME-2 model and measurement deviations of integrated system.

As shown in Fig. 6b, the contribution change of $PM_{2.5}$ factors with the change of total $PM_{2.5}$ concentration was analyzed. Secondary sulfate and biomass burning decreased when pollution was heavy while the secondary nitrate and SOA increased. Therefore, secondary conversion of carbon-containing and NOx components were the causes of heavy $PM_{2.5}$ pollution during

the observation period.

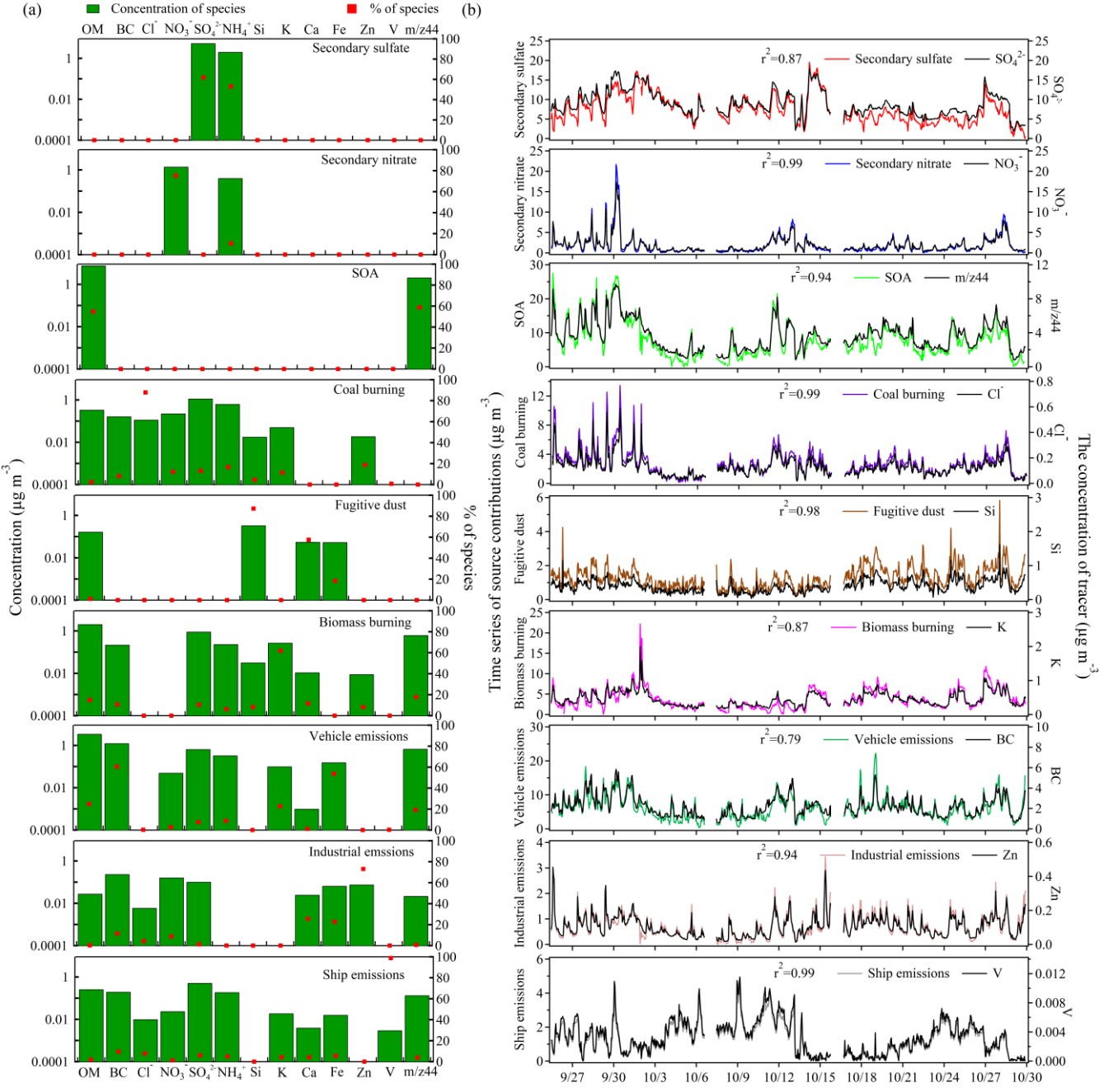

**Figure 5. Factor profiles and EV in the ME-2 model results(a). Time series of PM₂.₅ sources and tracers (b).**





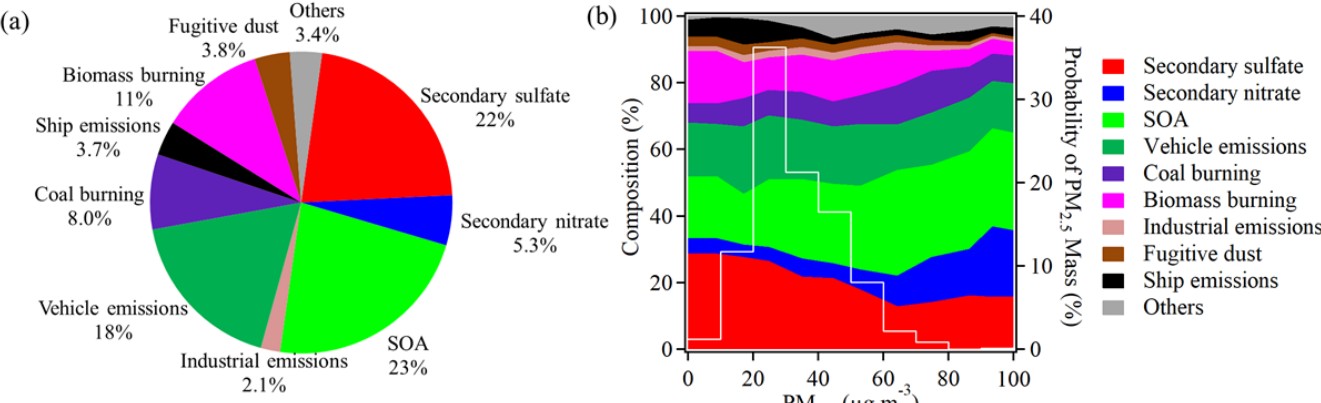


**Figure 6. Average contributions of each source to PM$_{2.5}$(a). Variation in the proportion of sources in PM$_{2.5}$ during the observation period. The white curve represents the occurrence probability of the PM$_{2.5}$ concentration (b).**

### 4.4.2 Diurnal variations

Diurnal variations of source were obtained based on the high time-resolution data, which could help to evaluate the validity of
the source apportionment and investigate the causes of PM$_{2.5}$ pollution further. The average diurnal trends of major sources
are shown in Fig. 7. Secondary sulfate diurnal variations presented relative high afternoon concentrations and low night-time
concentrations; moreover, a broad peak from 12:00 to 20:00 might be related to light intensity, suggesting that sulfate was
largely formed by photochemical processes (Liu et al., 2008). In contrast, secondary nitrate had a peak during 9:00-10:00 that
was slightly later than the morning rush hour for motor vehicles because the NOx in vehicle exhaust was an important source
of secondary nitrate. The pattern (high night-time concentration and low afternoon concentration) might be caused by the
diurnal variation of boundary layer. Decreasing in solar activity at night resulted in the height of boundary layer decreased
and the inversion layer occurred. (Xu et al., 2014), which was not conducive to the diffusion of pollutants. Moreover, the
pattern might due to the evaporation of ammonium nitrate produced by the secondary reaction at high temperature in the
afternoon. The variation of SOA was related to the photochemical reaction of VOCs. When the sun rises during the day, the
boundary layer was gradually elevated, which led to lower concentrations of residual SOA in the atmosphere, and it reached
its lowest point between 8:00 and 9:00. After that, SOA rose and peaked at 14:00 due to the enhancement of light. It was
consistent with the theoretical changes in SOA (Liu et al., 2018). For primary sources, vehicles, ships and industrial emissions
showed the characteristics of "two peaks and one valley", which were basically consistent with the law of human activities.
This finding suggested that human activities have a significant impact on environmental pollution. Biomass burning and
fugitive dust showed a trend of high day concentration and low night-time, which might be because of the local emissions
(such as biomass boiler, construction activities) during daytime that are mixed with the pollutants transported from other
regions as the rise of boundary layer at morning (Sun et al., 2019). The small peak at 2:00 was mostly related to the local
biomass burning activities at night. Diurnal variations of each source were more consistent with the actual situation, which
indicated that the results of source apportionment were reasonable.


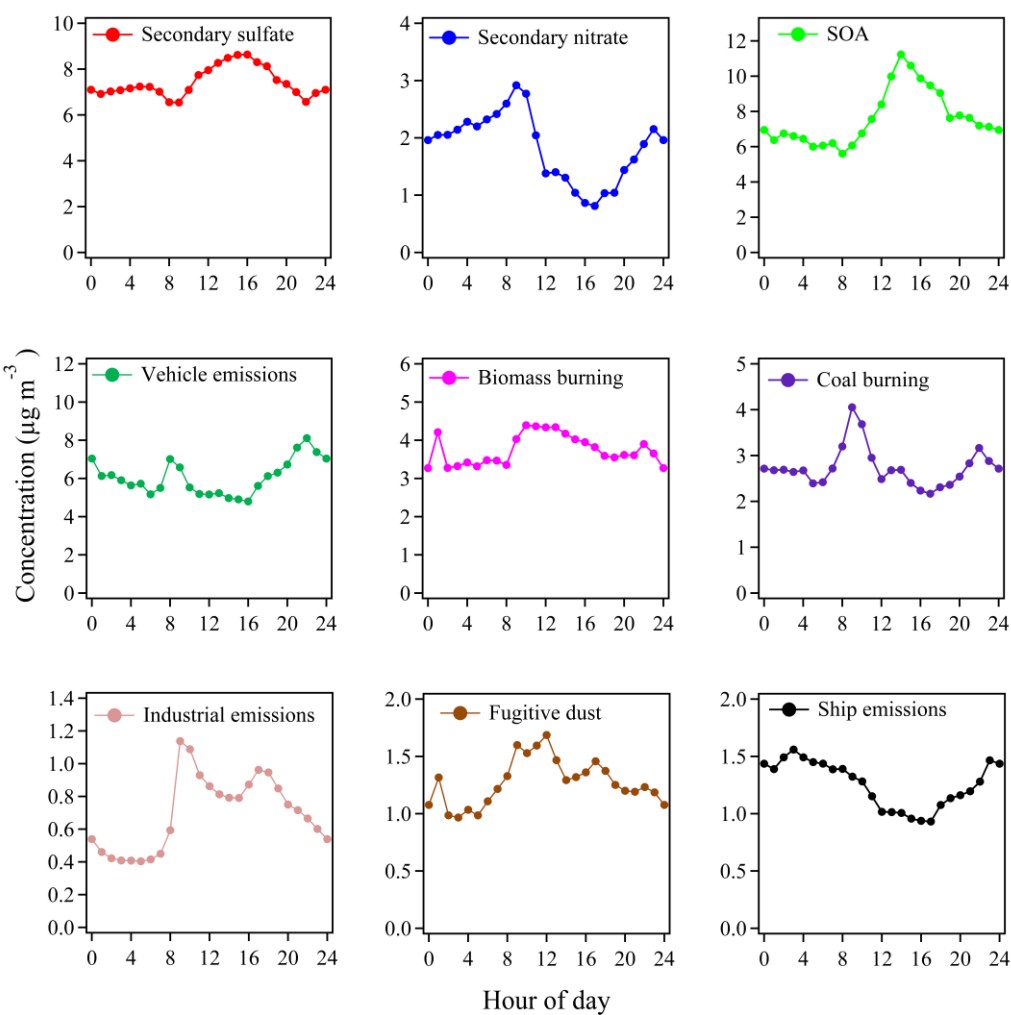

**Figure 7. Diurnal variations of the nine sources resolved by ME-2.**

### 4.4.3 Potential source areas

The potential source areas for different sources were identified by the PSCF model as shown in Fig. 8. During the observational
campaign, Shenzhen City mainly influenced by the continental and coastal air masses coming from the northeast of the domain
and to a lesser extent by the air masses originating from southwest of the city. The continental air masses transported from the
northern of Guangdong Province, Jiangxi Province, and Yangtze River Delta region, where there are highly urbanized and
industrialized areas. The coastal air masses stemming from the East Sea and Taiwan Strait. The secondary sulfate showed two
potentially important source areas located in the northern and southwestern of the sampling site. The southwestern potential
source areas indicated that regional influences were important to the secondary sulfate. Compared with secondary sulfate, the
high potential source region of secondary nitrate was closer to the sampling site and mainly located in the southwestern and





northern areas. These findings implied that secondary nitrate was from local secondary aerosol rather than regional transport. The PSCF plot of SOA displayed a clear high-potential source region in northern Guangdong Province, including the cities of Dongguan, Huizhou, and Shaoguan. The PSCF plot of vehicle emissions showed potential source areas around the sampling

site. Fugitive dust and biomass burning have similar potential source region originated from the northeast area of the site, suggesting the two sources may be influenced by the regional transport. This result was consistent with the high fugitive dust and biomass burning contributions to $PM_{2.5}$ at daytime, which are partly related to regional transport (Fig. 7). Sun et al. (2019) conducted vertical observations of air pollutants ($SO_2$, $O_3$, $NO_x$, $PM_{10}$, $PM_{2.5}$) and revealed the obvious regional transport in Shenzhen. In addition, biomass emission also showed high-potential source regions in the northern area Guangdong Province.

The PSCF plot of biomass burning indicates local emissions as well as regional influences. Coal burning high-potential source areas were similar but smaller compared to those of industrial emissions, which were manly located in the northern area of Shenzhen. The potential areas of ship emissions were consistent with the coastal regions, where there are many shipping ports, especially the Shenzhen ports, which are the third largest in the world and have active shipping lanes with traffic volumes up to 500,000 transits in 2015 (Mou et al., 2019). In summary, the PSCF analysis indicated that the northern domain was the high

potential source region for secondary aerosol, coal burning, and industrial emissions. Vehicle emissions and secondary nitrate were mostly influenced by local emissions. For fugitive dust, biomass burning, ship emissions, and secondary sulfates, the potential source areas on a regional scale were important in this study.

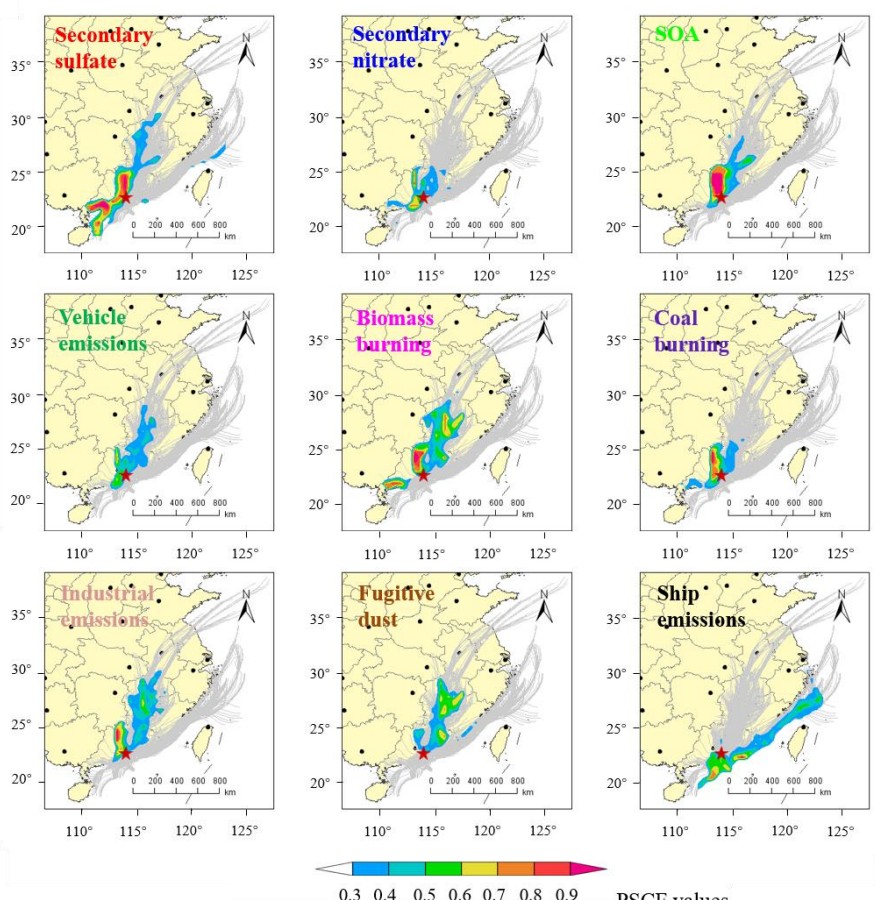

**Figure 8. Spatial distributions of PSCF values for the nine sources resolved by ME-2.**

## 5 Conclusions

In this work, a new online integrated system for the measurement of chemical species in $PM_{2.5}$ was developed by combining ACSM, Xact-625, AE-31, and SHARP-5030i. A sampling tube was designed to reduce aerosol transmission loss. Approximately 94% of $PM_{2.5}$ components can be detected, and the data (except $Cl^-$) were well correlated with those measured by other independent online instruments (MARGA, AE-31), with $r^2$ values ranging from 0.74 to 0.99. The concentrations of chemical components were similar to those of offline sampling. To more accurately reveal the $PM_{2.5}$ pollution characteristics, the new online integrated system was applied at the university area of Shenzhen in the autumn of 2019. The average $PM_{2.5}$ concentration was 33 μg m$^{-3}$ during the sampling period, of which $SO_4^{2-}$, OM, $NH_4^+$, BC, $NO_3^-$, important elements, and $Cl^-$ contributed 25%, 41%, 11%, 6.4%, 5.3%, 5.0%, and 0.38%, respectively. The chemical composition measured was close to the mass closure.



An improved source apportionment method was established by employing the ME-2 model on the mass spectra information (m/z44) and chemical species measured from the online integrated system to solve the SOA contributions. SOA was found to be the dominant source (23%), followed by secondary sulfate (22%), vehicle emissions (18%), biomass burning (11%), coal burning (8.0%), secondary nitrate (5.3%), fugitive dust (3.8%), ship emissions (3.7%), and industrial emissions (2.1%). The online integrated systems applied for source apportionment could obtain more accurate results through the analysis of time

series trend and diurnal variation. The emission source had high correlations with the tracer species ($r^2$=0.79-0.99). Secondary sulfate and SOA showed high concentrations during the daytime, which was associated with photochemical processes. The concentration of secondary nitrate was high at nighttime and low in the afternoon, and the variation of primary sources was basically consistent with the law of human activities, indicating that the results of source apportionment were reasonable. The results of the PCSF model indicated that the northern area of the sampling site was the potential source region during the

sampling campaign and regional transmission (e.g., biomass burning, fugitive dust, etc.) and played an important role in $PM_{2.5}$ pollution in this work. The development and successful application of the online integrated system and source apportionment method suggested that they can be used for precise regulation of $PM_{2.5}$.

*Data availability*. Datasets are available by contacting the corresponding author, Xiao-Feng Huang (huangxf@pku.edu.cn).


*Author contributions*. XFH, CPS, and XP analyzed the data and wrote the paper. XFH, LWZ, and LYH designed the study. CPS, MXT, YC and YW performed the chemical analysis. BZ performed the PSCF modelling. All authors reviewed and commented on the paper.

*Competing interests*. The authors declare that they have no conflict of interest.

*Acknowledgements.* This work was supported by the National Natural Science Foundation of China (91744202). We thank André S. H. Prévôt of the Paul Scherrer Institute, Switzerland, for providing the SoFi code of source apportionment.




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
