# Peer review of "Development and application of a mass closure PM2.5 composition online monitoring system"

_Atmospheric Measurement Techniques, 2020_

## Referee Comment (RC1) · Anonymous Referee #1 · 14 Apr 2020

General Comments: The manuscript by Su et al. developed a mass closure PM2.5 on-line integrated system and characterized the PM2.5 sources and composition in Shen-zhen. One of the major concerns of this study is that, apart from sharing a sampling line, what is the difference between using them separately and using the sampling system including aerosol chemical speciation monitor (Aerodyne, ACSM), Aethalometer (Magee, AE-31), automated multimetals monitor (Cooper Corporation, Xact-625), and hybrid synchronized ambient particulate real-time analyzer monitor (Thermo Scientific, SHARP-5030i) ? What is the purpose and significance of establishing such a sampling system given that I already have these instruments? On the other hand, the manuscript fails to identify the new information and seems to be a report regarding sources and composition of PM2.5 in a different location and season. Before its publication, the

following comments need to be addressed.

Specific Comments: 1. What is the difference between the sampling system and the separate detection by separate instrument?

2. What is the uncertainty quantification of chemical species measured by ACSM, AE31 and Xact-625? And what's the uncertainty quantification of PM2.5? What's the error margin of PM2.5 mass closure? What's the Detection Limit of each species?

3. More needs to be listed to support source apportionment results. For example, please make clear that Chl shows dominated contribution to coal combustion factor while negligible fraction in biomass burning. Besides the high mass loading of K in factor 6, is there any other evidences to support that factor6 is related to the biomass burning? In addition, the factor related coal combustion shows 3 peaks during daytime. Please explain this.

4. Although there is tight correlation between reconstructed and measured PM2.5, what are the reasons caused the underestimation in 10/20 and overestimation in 10/3? Please elaborate.

5. What ACSM is used in your manuscript? Q-ACSM or ToF-ACSM? Did you measure the PM1 species in this study (based on the reference of Nga et al., 2011)? if so, how about the gap of chemical species between the PM1 and PM2.5? Please mention it here. Also, more details in concentration and composition of PM species need to be shown wherever in the main text or the supplementary.

6. The frequency of data is negligible at PM2.5 > 80 $\mu$g/m3 in Fig.6. In my viewpoint, there might be a significant uncertainty in fraction of composition when PM levels are low. The authors need to address such uncertainties in the revised manuscript.

7. Line 31: "PM2.5 is currently a serious problem" is not suitable expression.

Please also note the supplement to this comment:

[Figure]

https://www.atmos-meas-tech-discuss.net/amt-2020-77/amt-2020-77-RC1-supplement.pdf

[Figure]

**Supplement:**

General Comments:

The manuscript by Su et al. developed a mass closure $PM_{2.5}$ online integrated system and characterized the $PM_{2.5}$ sources and composition in Shenzhen. One of the major concerns of this study is that, apart from sharing a sampling line, what is the difference between using them separately and using the sampling system including aerosol chemical speciation monitor (Aerodyne, ACSM), Aethalometer (Magee, AE-31), automated multimetals monitor (Cooper Corporation, Xact-625), and hybrid synchronized ambient particulate real-time analyzer monitor (Thermo Scientific, SHARP-5030i) ? What is the purpose and significance of establishing such a sampling system given that I already have these instruments? On the other hand, the manuscript fails to identify the new information and seems to be a report regarding sources and composition of $PM_{2.5}$ in a different location and season. Before its publication, the following comments need to be addressed.

Specific Comments:

1. What is the difference between the sampling system and the separate detection by separate instrument?

2.  What is the uncertainty quantification of chemical species measured by ACSM, AE31 and Xact-625? And what's the uncertainty quantification of $PM_{2.5}$? What's the error margin of $PM_{2.5}$ mass closure? What's the Detection Limit of each species?

3. More needs to be listed to support source apportionment results. For example, please make clear that Chl shows dominated contribution to coal combustion factor while negligible fraction in biomass burning. Besides the high mass loading of K in factor 6, is there any other evidences to support that factor6 is related to the biomass burning? In addition, the factor related coal combustion shows 3 peaks during daytime. Please explain this.

4. Although there is tight correlation between reconstructed and measured $PM_{2.5}$, what are the reasons caused the underestimation in 10/20 and overestimation in 10/3? Please elaborate.

5. What ACSM is used in your manuscript? Q-ACSM or ToF-ACSM? Did you measure the $PM_1$ species in this study (based on the reference of Nga et al., 2011)? if so, how about the gap of chemical species between the $PM_1$ and $PM_{2.5}$? Please mention it here. Also, more details in concentration and composition of PM species need to be shown wherever in the main text or the supplementary.

6. The frequency of data is negligible at $PM_{2.5} > 80$ μg/m$^3$ in Fig.6. In my viewpoint, there might be a significant uncertainty in fraction of composition when PM levels are low. The authors need to address such uncertainties in the revised manuscript.

7. Line 31: "$PM_{2.5}$ is currently a serious problem" is not suitable expression.

---

## Referee Comment (RC2) · Anonymous Referee #2 · 13 Jul 2020

This paper reports the development of an online monitoring system for PM2.5 composition. The unique feature for this system is the capability to measure more than 90% of PM2.5 mass with a high time resolution. This advance makes this system a powerful tool for understanding PM2.5 sources and deciding the corresponding control measures. Another novelty of this paper is the resolving of secondary organic aerosol (SOA) from the total PM2.5 mass by the usage of m/z 44, a good SOA tracer, in the PMF modelling. This system has been successfully applied in a megacity in China, with nine sources well identified, supporting its effectiveness and usefulness in PM2.5 control. Overall, I think it is a well-written paper with novelty and recommend its publication after considering the following concerns.

1. The author should make a clearer statement of the advantage of the new system

compared to the separate instruments in the introduction part.

2. Line 31. "serious" might not be suitable to the current PM2.5 pollution status. A description of long-term problem should be better.

3. Why Na was not measured? Could sea salt be a major source for PM2.5 in Shenzhen, a coastal city?

4. 2.4 Design of the data analytics platform. More details for data conversion of each instrument should be given.

5. Figure 2a. The mass closure is generally good. However, the authors should comment on some periods when a significant mass discrepancy appeared.

6. Figure 4. The elements with low concentrations should be displayed with enlargement.

7. Figure 7. There is a spike at 1:00 am for both biomass burning and fugitive dust. Why?

8. Conclusions. For a technical paper, the prospects of further development of the new system should be given.

---

## Author Comment (AC1) · 27 Jul 2020

We want to thank the reviewers for their helpful comments and suggestions which have helped to improve our manuscript. We have provided a point by point response to the comments (see below), and have modified our manuscript accordingly.

**Review 1:**

General Comments: The manuscript by Su et al. developed a mass closure $PM_{2.5}$ online integrated system and characterized the $PM_{2.5}$ sources and composition in Shenzhen. One of the major concerns of this study is that, apart from sharing a sampling line, what is the difference between using them separately and using the sampling system including aerosol chemical speciation monitor (Aerodyne, ACSM), Aethalometer (Magee, AE-31), automated multimetals monitor (Cooper Corporation, Xact-625), and hybrid synchronized ambient particulate real-time analyzer monitor (Thermo Scientific, SHARP-5030i)? What is the purpose and significance of establishing such a sampling system given that I already have these instruments? On the other hand, the manuscript fails to identify the new information and seems to be a report regarding sources and composition of $PM_{2.5}$ in a different location and season. Before its publication, the following comments need to be addressed.

**Response:** We would like to thank the reviewer for the comments. Regarding the purpose and significance of this study, we would like to make the following explanation:

First, it is difficult for online instruments to sharing a sampling line, because the flow rate and measurement principle of the instruments are different. Reducing particulate matter loss, controlling temperature and humidity during the sampling process is also important. Therefore, unlike independent sampling, we have spent a lot of effort to achieve integrated sampling using isokinetic sampling manifold.

Second, as the reviewer mentioned, we already have these instruments. However, sampling separately cannot achieve $PM_{2.5}$ mass closure and source analysis, so we tried to integrate them. The dataset obtained is comparable and more reliable.

Third, the choice of instruments in the integrated system is not involved in other studies. ACSM was integrated into the new system to measure OM and make PM$_{2.5}$ mass closure better.

Last, it is not only a report of source analysis, we used the data of the integrated system to improve the time resolution of the source analysis results, and resolved the secondary organic aerosol (SOA) from the total PM$_{2.5}$ mass by the usage of m/z 44 (a good tracer). This research had made a demonstration for PM$_{2.5}$ full component monitoring and source analysis.

We explained the above content in detail and added it in the manuscript when addressed the major comments.

Major comments

1. What is the difference between the sampling system and the separate detection by separate instrument?

**Response**: In previous studies, separate instruments were widely used to measure the chemical composition of a certain class, but cannot achieve the quantification of all or most of PM$_{2.5}$ mass concentration. Compared with separate instruments, in our study, the same sampling head was used to collect particles, and then divides into four monitoring instruments to monitor the corresponding components. The sampling system aimed to ensure the reliability and comparability of synchronous sampling of different instruments, reduce the different influence of loss of drainage pipes, temperature, humidity on different instruments, and output unified PM$_{2.5}$ component data that can be directly applied to source analysis.

To illustrate this point, we made the following changes in the original manuscript:

"There are two differences between the new online integrated system and the separate online instruments. On one hand, the new online integrated system used isokinetic sampling manifold and the same sampling head to ensure the reliability and

comparability of synchronous sampling among different instruments. On the other hand, ACSM was integrated into the new system to measure OM and make PM$_{2.5}$ mass closure better." (Line 62-66)

2. What is the uncertainty quantification of chemical species measured by ACSM, AE31 and Xact-625? And what's the uncertainty quantification of PM$_{2.5}$? What's the error margin of PM$_{2.5}$ mass closure? What's the Detection Limit of each species?

**Response**: We thank the reviewer for this question, we added uncertainty quantification of each instruments in the supplementary material, and hoped those can give reader a better idea about the reliability of integrated systems. we also have added the table about Detection Limit to SI (in Table S1):

"Uncertainty analysis on mass concentrations of integrated system

For Xact-625, the combined uncertainty included contribution from flow (1.5%), calibration standard uncertainty (5%), long term stability (calculated from the standard deviation of hourly internal Pd reference, 1.3%), and an element-specific uncertainty associated with the spectral deconvolution calculated by instrument software for each spectrum (US-EPA,1999; Tremper et al.,2018). In our study, each of elements was calibrate individually with a reference sample, the Xact-625 LOD (Limit of detection) was calculated using HEPA field blank measurements during sampling campaign these are shown in Table S1.

The uncertainty of ACSM is similar to AMS, as described previously (Allan et al.,2003; Nga et al., 2011), the overall uncertainty includes uncertainties associated with the Q (flow), IE (Ionization Efficiencies), RIE (Relative Ionization Efficiencies), and CE (Collection Efficiency) (Middlebrook et al., 2012; Freney et al., 2019). In which, IE is the ionization, transmission, and ion detection efficiency of nitrate (typically shortened as "ionization efficiency"), 10% uncertainty. In this study, the capture vaporizer (CV) was equipped, compared with a standard vaporizer (SV), to reduce the particle bouncing effect at vaporizer and the particle bouncing effect at vaporizer and hence improves the quantitative uncertainties caused by CE (Hu et al., 2017b; Zhang et al.,

2017), the uncertainty in CE is less than 30%. RIE is the ionization efficiency of species relative to the ionization efficiency of nitrate; for ammonium and sulfate: determined in routine calibrations (10% uncertainty and 30% uncertainty, respectively); for organics: determined for various types of organics in previous laboratory experiments and literature values, an average value is used (20% uncertainty) (Bahreini et al., 2009). Q is the volumetric sample flow rate into the instrument (<0.5% uncertainty). The propagated, overall uncertainty for the total ACSM mass concentration is 20%-30%.

For AE-31 and SHARP-5030i, according to instrument manufacturer's test of instruments, the measurement accuracy is 5%, that is, the measurement error is within 5% of the measured value.

In addition, to guarantee the data quality acquired by the integrated system, relevant quality assurance and quality control (QA/QC) measures are implemented. The calibration of sampling flow rate, blank experiment and instrument calibration were performed periodically to ensure data quality according to relevant national standards. The sampling flow rate was calibrated every month to ensure the sampling flow precision was in the range of ±1.5%. The blank experiment and instrument calibration were conducted every two months."

"Table S1. The concentrations and detection limit of $PM_{2.5}$ and chemical species during the sampling campaign."

| | Species | Average concentration | Standard deviation | Detection Limit |
|---|---|---|---|---|
| Organic ($\mu g\ m^{-3}$) | OM | 14.1 | 7.4 | 0.3 |
| Inorganic ions ($\mu g\ m^{-3}$) | $SO_4^{2-}$ | 8.6 | 3.3 | 0.4 |
| | $NO_3^-$ | 1.8 | 1.9 | 0.2 |
| | $NH_4^+$ | 3.8 | 1.7 | 0.5 |
| | $Cl^-$ | 0.1 | 0.07 | 0.2 |
| | BC | 2.1 | 1.0 | 0.1 |
| Trace elements ($ng\ m^{-3}$) | Si | 380.6 | 185.0 | 17.8 |
| | K | 443.9 | 269.1 | 1.17 |
| | Ca | 103.0 | 53.8 | 0.3 |
| | Ti | 14.4 | 8.2 | 0.16 |
| | V | 3.2 | 2.3 | 0.12 |
| | Cr | 2.9 | 2.0 | 0.12 |

| | | | |
|---|---|---|---|
| Mn | 24.3 | 13.0 | 0.14 |
| Fe | 288.7 | 132.2 | 0.17 |
| Co | 0.03 | 0.1 | 0.14 |
| Ni | 2.9 | 1.3 | 0.1 |
| Cu | 11.3 | 7.7 | 0.079 |
| Zn | 102.2 | 60.9 | 0.067 |
| As | 5.8 | 4.7 | 0.063 |
| Se | 2.2 | 1.2 | 0.081 |
| Mo | 0.5 | 0.5 | 0.1 |
| Cd | 7.3 | 3.2 | 2.5 |
| Sn | 19.8 | 8.3 | 4.1 |
| Sb | 28.0 | 10.2 | 5.2 |
| Ba | 3.9 | 7.4 | 0.39 |
| Hg | 1.9 | 0.7 | 0.12 |
| Pb | 18.6 | 9.5 | 0.13 |

The integrated system has a small error margin of mass closure, which is one of the advantages of the system. We explained in the manuscript:

"The average error margin of mass closure during the observation period is about 6%, which might be due to the measurement error of the integrated instruments, particle composition, temperature and relative humidity (Su et al., 2018; Zhang et al., 2017)." (Line 227-228).

3. More needs to be listed to support source apportionment results. For example, please make clear that Chl shows dominated contribution to coal combustion factor while negligible fraction in biomass burning. Besides the high mass loading of K in factor 6, is there any other evidences to support that factor6 is related to the biomass burning? In addition, the factor related coal combustion shows 3 peaks during daytime. Please explain this.

**Response**: Thank you for your questions about coal and biomass burning, which reminded us to think deeply about these two sources. $Cl^-$ in $PM_{2.5}$ originates mainly from ammonium chloride, which is formed by the rapid combination of HCl and

ammonia in the atmosphere. In China, a country with huge coal-consumption, the biggest potential source of HCl is coal burning (Yudovich and Ketris, 2006; Wang et al., 2015). Huang et al. (2018) found that the concentration distribution of $Cl^-$ in the Pearl River Delta is consistent with the distribution of coal-fired power plants in the source list. The factor also had some mass contributions from $SO_4^{2-}$, $NO_3^-$, and EC, further confirming that it was from coal burning (Zheng et al.,2013).

Factor 6 was identified as biomass combustion not only because that K is a tracer for biomass burning, but also for the following reasons. First, the factor had a large part of mass contribution of OM, and biomass burning is a significant source of $PM_{2.5}$ and major organic carbon in China (Akagi et al. 2011). Second, the factor had a certain amount of Zn, open-air biomass burning often mixes with garbage burning in Chinese rural areas. The smoke of garbage-burning emission might contain Zn (Zou et al.,2017). Third, according to previous study about source inventory surveys, particulate matter emitted from biomass combustion usually contains a certain amount of Ca (Ma et al., 2015). Besides, as Fig.8 shown, biomass burning has a high contribution from north and south-west, which is in agreement with the distribution of fire points in Guangdong (Fig.S6), it can be clearly seen that Shenzhen surrounding areas present a high-density fire point distribution, indicating frequent occurrence of biomass burning. The activities of biomass burning often occurs during the daytime, that is the reason why biomass burning had a trend of high day concentration and low night-time (Fig.7). Therefore, we think that factor 6 can be identified as biomass combustion.

The daily variation of coal burning sources derived from $Cl^-$ is consistent (Fig S7.), and there is a peak around 9:00 in the morning. It might because the temperature starts to rise in the morning and the vertical air convection activity is strengthened, bringing the pollutants transmitted by overhead sources to the near ground. As for the other two peaks, we found that it was the abnormally high contribution of coal-burning around 14:00 on September 25, so there might be local source that caused the peak of the diurnal variations. The spike around 22:00 was mainly because of China's National Day on October 1, there were displays of fireworks from September 25 to October 1 at 3

playgrounds 10 km away from sampling site. Previous study has shown that the concentration of species (e.g. K, Ca, $Cl^-$, $NO_3^-$) in the $PM_{2.5}$ would have greatly increase due to the fireworks (Tsai et al., 2012).

We also added more specifically state the sources in manuscript:

"Factor 4 was associated with coal burning due to the high EV value of $Cl^-$, which is formed by the rapid combination of HCl and ammonia in the atmosphere, and the potential source of HCl is coal burning in China (Yudovich and Ketris, 2006; Wang et al., 2015), and the concentration distribution of $Cl^-$ in the Pearl River Delta was consistent with the distribution of coal-fired power plants in the source list (Huang et al., 2018). This factor also had some mass contributions from $SO_4^{2-}$, $NO_3^-$, and EC, further confirming that it was from coal burning (Zheng et al.,2013)." (Line 289-293)

"Factor 6 has a high loading of K and a certain of OM and BC. K has been used as a clear tracer for biomass burning (Yamasoe et al., 2000; Sillapapiromsuk et al., 2013). Biomass burning is a significant source of $PM_{2.5}$ and organic matter in China (Akagi et al., 2011). The Zn contained in this factor might be related to garbage-burning (Yuan et al., 2006). Besides, biomass burning has a high contribution from north and south-west (Fig. 8), which is consistent with the distribution of fire points in Guangdong (Fig.S8). The activities of biomass burning often occurs during the daytime, which consisted with the diurnal pattern of biomass burning that high day concentration and low night-time (Fig. 7)." (Line 297-302)

"The daily variation of coal burning and $Cl^-$ was consistent (Fig S7), and there is a peak around 9:00 in the morning. It might because the temperature starts to rise in the morning and the vertical air convection activity is strengthened, bringing the pollutants transmitted by overhead sources to the near ground. The spike around 22:00 was mainly because of firework shows during September 25 to October 1. The peak around 14:00 mainly came from local sources on September 25." (Line 361-364)

[Figure]

Figure S6. Spatial distribution of fire points in Guangdong (divided into day and night) during observation (the five-pointed star was used to represent sampling site).

[Figure]

Figure S7. Diurnal variation of the Cl- during the observation.

4. Although there is tight correlation between reconstructed and measured PM2.5, what are the reasons caused the underestimation in 10/20 and overestimation in 10/3? Please elaborate.

**Response**: In the process of sampling, we tried our best to do a good job in quality control and guarantee. However, PM2.5 monitor instruments of different principles can be overestimated and underestimated for PM2.5 (Su et al., 2018; Zhang et al., 2017; Budisulistiorini et al., 2014), so are the reconstruction and measurement (Ji et al., 2018). We think it's almost inevitable, the possible reasons are as follows, the measurement error of the integrated instruments, particle composition, temperature and relative humidity and so on. In our study, the deviation between the reconstructed and measured PM2.5 form a normal distribution, ranging from -30% to +30%. On the whole, the correlation is tight and the slope is close to 1, which indicates that the data is relatively reasonable.

As the reviewer pointed out that the fitting problem of October 3 and October 20, we think it may be related to the temperature. The temperature on October 3 is the highest during the observation period, and at October 20, the temperature is at a relatively low level. The temperature has different effects on SHARP-5030i and integrated instruments, which results in different fitting effects to a certain extent. Of course, there

may be other reasons just as mentioned above.

We also pointed out the reasons caused the underestimation in the manuscript:

"The average error margin of mass closure during the observation period is about 6%, which might be due to the measurement error of the integrated instruments, particle composition, temperature and relative humidity (Su et al., 2018; Zhang et al., 2017). A significant mass discrepancy between reconstructed and measured $PM_{2.5}$ appeared in some periods (Fig. 2a). For example, the underestimation on October 3 and the overestimation on October 20 occurred when the temperature was the highest and the lowest during the observation period, respectively. Therefore, it was speculated that temperature might affect the composition of $PM_{2.5}$, causing the mass closure to deviate." (Line 227-232)

5. What ACSM is used in your manuscript? Q-ACSM or ToF-ACSM? Did you measure the PM1 species in this study (based on the reference of Nga et al., 2011)? if so, how about the gap of chemical species between the PM1 and $PM_{2.5}$? Please mention it here. Also, more details in concentration and composition of PM species need to be shown wherever in the main text or the supplementary.

**Response**: We are sorry for not describing it clearly, $PM_{2.5}$-Q-ACSM was used and the $PM_1$ species was not be measured in this study. But, the performance of $PM_{2.5}$-Q-ACSM have been fully evaluated in both laboratory and field studies (Hu et al., 2017a; Xu et al.,2017). The gap of chemical species between $PM_{2.5}$-Q-ACSM and $PM_1$-Q-ACSM also has been analyzed in previous study: the comparisons between the two Q-ACSMs illustrated similar temporal variations in all non-refractory (NR) species between $PM_1$ and $PM_{2.5}$($r^2$>0.9); one average, NR-$PM_{1-2.5}$ contributed 53% of the total NR-$PM_{2.5}$, the ratios of [NR-$PM_1$]/[NR-$PM_{2.5}$] varied among different species. In particular, nitrate and chloride showed much higher [NR-$PM_1$]/[NR-$PM_{2.5}$] ratios compared with other species (Zhang et al.,2017).

In the manuscript, we present the time series stack diagrams of several classes of components (Fig. 2(a)) and the proportions of all components (Fig. 3), and describe

them in main text (Line 214-220). In addition, we supplement the concentration and relative standard deviation of each component in SI (Table S1). To clarify this question, we have modified the description of the ACSM in the manuscript.

"PM$_{2.5}$-capable Q-ACSM was selected for monitoring the mass concentrations of ambient OM, SO$_4^{2-}$, NO$_3^-$, NH$_4^+$, and Cl$^-$ in non-refractory PM$_{2.5}$. The PM$_{2.5}$-Q-ACSM equipped with a PM$_{2.5}$ lens and capture vaporizer (CV), and can detect approximately 90% of PM$_{2.5}$ particles (Hu et al., 2017; Xu et al.,2017; Zhang et al., 2017)" (Line 95-97)

6. The frequency of data is negligible at PM$_{2.5}$ > 80μg/m$^3$ in Fig.6. In my view point, there might be a significant uncertainty in fraction of composition when PM levels are low. The authors need to address such uncertainties in the revised manuscript.

**Response**: Thank you for pointing it out. During our observation period, the proportion of data that PM$_{2.5}$>80μg/m$^3$ is only 1.3%, so we combined the data that PM$_{2.5}$ > 80 μg/m$^3$ into the concentrations of 70-80 μg/m$^3$. Fig. 6(b) has been revised.

[Figure]

Figure 6. Average contributions of each source to PM$_{2.5}$(a). Variation in the proportion of sources in PM$_{2.5}$ during the observation period. The white curve represents the occurrence probability of the PM$_{2.5}$ concentration (b). (Line 335)

The uncertainties of PM$_{2.5}$ and their compositions have been clarified in the secondary question and added into SI, when PM level were below the detection limit, we use the half detection limit (DL) instead the concentration, and their uncertainties was set as 5/6 DL when source resolution was calculated. Please see lines 171-174.

7. Line 31: "PM$_{2.5}$ is currently a serious problem" is not suitable expression

**Response**: Thank you for this comment. The sentence was not clear enough and could be misunderstood. Therefore, we changed it to: "PM$_{2.5}$ is a long-term problem in some cities or regions." (Line 31)

**Reference**

Akagi, S. K., Yokelson, R. J., Wiedinmyer, C., Alvarado, M. J., Reid, J.S., Karl, T., Crounse, J. D., Wennberg, P.: Emission factors for open and domestic biomass burning for use in atmospheric models., Atmos. Chem. Phys., 11, 4039-4072, http://doi.org/10.5194/acpd-10-27523-2010, 2011.

Allan, J. D., Jimenez, J. L., Williams, P. I., Alfarra, M. R., Bower, K. N., Jayne, J. T., Coe, H., and Worsnop, D. R.: Quantitative sampling using an Aerodyne aerosol mass spectrometer 1. Techniques of data interpretation and error analysis, J. Geophys.Res.,108(D3),4090, http://doi.org/10.1029/2002jd002358, 2003.

Bahreini, R., Ervens, B., Middlebrook, A. M., Warneke, C., de Gouw, J. A., Decarlo, P. F., Jimenez, J. L., Brock, C. A.,Neuman, J. A., Byerson, T. B., Stark, H., Atlas, E., Brioude, J., Fried, A., Holloway, J. S., Peischl, J., Richter, D., Walega, J., Weibring, P., Wollny, A. G., and Fehsenfeld, F. C.: Organic aerosol formation in urban and industrial plumes near Houston and Dallas, Texas., J. Geophys. Res., VOL.114, D00F16, http://doi.org/10.1029/2008JD011493, 2009.

Budisulistiorini, S. H., Canagaratna, M. R. Croteau, P. L., Baumann, K., Edgerton, E. S., Kollman, M. S., Ng, N. L., Verma, V., Shaw, S. L., Knipping, E. M., Worsnop, D. R., Jayne, J. T., Weber, R. J., and Surratt, J. D.: Intercomparison of an Aerosol Chemical Speciation Monitor (ACSM) with ambient fine aerosol measurements in downtown Atlanta, Georgia., Atmos. Meas. Tech., 7, 1929-1941, http://doi.org/10.5194/amt-7-1929-2014, 2014.

Freney, E., Zhang, Y. J., Croteau, P., Amodeo, T., Williams, L., Truong, F., Petit, J., Sciare, J., Sarda-Esteve, R., Bonnaire, N., Arumae, T., Aurela, M., Bougiatioti, A., Mihalopoulos, N., Coz, E., Artinano, B., Crenn, V., Elste, T., Heikkinen, L., Poulain, L., Wiedensohler, A., Herrmann, H., Priestman, M., Alastuey, A., Stavroulas, I., Tobler, A., Vasilescu, J., Zanca, N., Canagaratna, M., Carbone, C., Flentje, F., Green, D., Maasikmets, M., Marmureanu, L., Minguillon, M. C., Prevot, A. S. H., Gros, V., Jayne, J., and Favez, O.: The second ACTRIS intercomparison (2016) for Aerosol Chemical Speciation Monitors (ACSM): Calibration protocols and instrument performance evaluations, Aerosol Science and Technology, 53(7), 830-842, http://doi.org/10.1080/02786826.2019.1608901., 2019.

Hu, W. W., Campuzano-Jost, P., Day, D. A., Croteau, P., Canagaratna, M. R., Jayne, J. T., Worsnop, D. R., and Jimenez, J. L.: Evaluation of the new capture vapourizer for aerosol mass spectrometers (AMS) through laboratory studies of inorganic species, Atmos. Meas. Tech., 10, 2897-2921, https://doi.org/10.5194/amt-10-2897-2017, 2017a.

Hu, W. W., Campuzano-Jost, P., Day, D. A., Croteau, P., Canagaratna, M. R., Jayne, J.T., Worsnop, D.R., and Jimenez, J. L.: Evaluation of the new capture vaporizer for aerosol mass spectrometers (AMS) through field studies of inorganic species,

Aerosol Science and Technology, 51(6), 735-754, http://doi.org/10.1080/02786826.2017.1296104, 2017b.

Huang, X. F., Zou, B. B., He, L. Y., Hu, M., Prévôt, A.S.H., and Zhang, Y. H.: Exploration of PM2:5 sources on the regional scale in the Pearl River Delta based on ME-2 modeling. Atmos. Chem. Phys., 18(16), 11563-11580, http://doi.org/10.5194/acp-18-11563-2018, 2018.

Ji, D. S., Cui, Y., Li, L., He, J., Wang, L. L., Zhang, H. L., Wang, W., Zhou, L. X., Maenhaut, W., Wen, T. X., Wang, Y. S.: Characterization and source identification of fine particulate matter in urban Beijing during the 2015 Spring Festival, Sci. Total. Environ., 628-629:430-440, http://doi.org/ 10.1016/j.scitotenv.2018.01.304, 2018.

Ma, Z. H., Liang, Y. P., Zhang, J., Zhang, D. W., Shi, A. J., Hu, J. N., Lin, A. G., Feng, Y. J., Hu, Y. Q., and Liu, B.X.: PM$_{2.5}$ profiles of typical sources in Beijing.Acta Scientiae Circumstantiae, 35(12): 4043-4052, http://doi.org.10.13671/j.hjkxxb.2015.0584, 2015.

Middlebrook, A. M., Bahreini, R., Jimenez, J. L., and Canagaratna, M. R.: Evaluation of Composition-Dependent Collection Efficiencies for the Aerodyne Aerosol Mass Spectrometer using Field Data., Aerosol.Sci.Tech., http://doi.org/10.1080/02786826.2011.620041, 46(3):258-271, 2012.

Nga, N. L., Herndona, S. C., Trimborna, A., Canagaratnaa, M. R., Croteaua, P. L., Onascha, T. B., Sueperab, D., Worsnopa, D. R., Zhang, Q., Sun, Y. L., and Jayne, J. T.: An Aerosol Chemical Speciation Monitor (ACSM) for Routine Monitoring of the Composition and Mass Concentrations of Ambient Aerosol, Atmos. Chem. Phys., 45, 770-784, http://doi.org/10.1080/02786826.2011.560211, 2011.

Sillapapiromsuk, S., Chantara, S., Tengjaroenkul, U., Prasitwattanaseree, S., Prapamontol, T.: Determination of PM10 and its ion composition emitted from biomass burning in the chamber for estimation of open burning emissions. Chemosphere. 93, 1912-1919, 2013.

Su, Y. S., Sofowote, U., Debosz, J., White, L., and Munoz, A.: Multi-Year Continuous PM$_{2.5}$ Measurements with the Federal Equivalent Method SHARP 5030 and Comparisons to Filter-Based and TEOM Measurements in Ontario, Canada, Atmosphere, 9(5), 191, https://doi.org/10.3390/atmos9050191, 2018.

Tremper, A. H., Font, A., Priestman, M., Hamad, S. H., Chung, T. C., Pribadi, A., Brown, R. J.C., Goddard, S. L., Grassineau, N., Petterson, K., Kelly, F. J., and Green, D.C.: Field and laboratory evaluation of a high time resolution x-ray fluorescence instrument for determining the elemental composition of ambient aerosols, Atmos. Meas. Tech., 11, 3541-3557, https://doi.org/10.5194/amt-11-3541-2018, 2018.

Tsai, H. H., Chien, L. H., Yuan, C. S., Lin, Y. C., Jen, Y. H., Ie, I. R.: Influences of fireworks on chemical characteristics of atmospheric fine and coarse particles during Taiwan's Lantern Festival, Atmospheric Environment., 62(2012), 256-264,

http://doi.org/10.1016/j.atmosenv.2012.08.012, 2012.

US-EPA: Determination of metals in ambient particulate matter using X-Ray Fluorescence (XRF) Spectroscopy, Agency, edited by: US-EPA (US Environmental Protection Agency), Cincinnati, OH 45268, USA, 1999.

Wang, Q.Q., Huang, X.H.H., Zhang, T., Zhang, Q. Y., Feng, Y. M., Yuan, Z. B., Wu, D., Lau, A.K.H., and Yu, J.Z.: Organic tracer-based source analysis of $PM_{2.5}$ organic and elemental carbon: a case study at Dongguan in the Pearl River Delta, China. Atmos. Environ. 118, 164-175, http://dx.doi.org/10.1016/j.atmosenv.2015.07.033, 2015.

Xu, W., Croteau, P., Williams, L., Canagaratna, M., Onasch, T., Cross, E., Zhang, X., Robinson, W., Worsnop, D., and Jayne, J.: Laboratory characterization of an aerosol chemical speciation monitor with $PM_{2.5}$ measurement capability, Aerosol Sci. Technol., 51(1), 69-83, https://doi.org/10.1080/02786826.2016.1241859, 2017.

Yamasoe, M. A., Artaxo, P., Miguel, A. H., and Allen, A. G.: Chemical composition of aerosol particles from direct emissions of vegetation fires in the Amazon Basin: water-soluble species and trace elements, Atmos. Environ., 34, 1641-1653, https://doi.org/10.1016/S1352-2310(99)00329-5, 2000.

Yuan, Z., Lau, A., Zhang, H., Yu, J., Louie, P., and Fung, J.: Identification and spatiotemporal variations of dominant PM10 sources over Hong Kong, Atmos. Environ., 40, 1803-1815, https://doi.org/10.1016/j.atmosenv.2005.11.030, 2006.

Yudovich, Y. E., and Ketris, M. P.: Chlorine in coal: a review., Int. J. Coal Geol., 67,127-144, https://doi.org/10.1016/j.coal.2005.09.004, 2006.

Zhang, Y. J., Tang, L.L., Croteau, P. L., Favez, O., Sun, Y., Canagaratna, M. R., Wang, Z., Couvidat, F., Albinet, A., Zhang, H. L., Sciare, J., Prévôt, A.S.H., Jayne, J.T., and Worsnop, D. R.: Field characterization of the $PM_{2.5}$ Aerosol Chemical Speciation Monitor: insights into the composition, sources, and processes of fine particles in eastern China., Atmos. Chem. Phys., 17, 14501-14517, https://doi.org/10.5194/acp-17-14501-2017, 2017.

Zheng, M., Zhang, Y. J., Yan, C.Q., Fu, H.Y., Niu, H.Y., Huang, K., Hu, M., Zeng, L.M., Liu, Q. Z., Pei, B., Fu, Q.Y.: Establishing $PM_{2.5}$ industrial source profiles in Shanghai., China Environmental Science, 33(8): 1354~1359, http://doi.org/CNKI:SUN:ZGHJ.0.2013-08-002, 2013.

Zou, B. B., Huang, X. F., Zhang, B., Dai, J., Zeng, L. W., Feng, N., and He, L. Y.: Source apportionment of $PM_{2.5}$ pollution in an industrial city in southern China., Atmos. Pollut. Res., 8(6), 1193-1202, https://doi.org/10.1016/j.apr.2017.05.001, 2017.

---

## Author Comment (AC2) · 27 Jul 2020

**Review 2:**

This paper reports the development of an online monitoring system for PM$_{2.5}$ composition. The unique feature for this system is the capability to measure more than 90% of PM$_{2.5}$ mass with a high time resolution. This advance makes this system a powerful tool for understanding PM$_{2.5}$ sources and deciding the corresponding control measures. Another novelty of this paper is the resolving of secondary organic aerosol (SOA) from the total PM$_{2.5}$ mass by the usage of m/z 44, a good SOA tracer, in the PMF modelling. This system has been successfully applied in a megacity in China, with nine sources well identified, supporting its effectiveness and usefulness in PM$_{2.5}$ control. Overall, I think it is a well-written paper with novelty and recommend its publication after considering the following concerns.

**Response:** Thank you.

**Comments:**

1. The author should make a clearer statement of the advantage of the new system compared to the separate instruments in the introduction part.

**Response**: Thanks for your comment. The statement of the new system has been added to the main text:

"There are two differences between the new online integrated system and the separate online instruments. On one hand, the new online integrated system used isokinetic sampling manifold and the same sampling head to ensure the reliability and comparability of synchronous sampling among different instruments. On the other hand, we integrated ACSM into the new system to measure OM to achieve PM$_{2.5}$ mass closure better" (Line 62-66)

2. Line 31. "serious" might not be suitable to the current PM$_{2.5}$ pollution status. A description of long-term problem should be better.

**Response**: Thank you for this comment and suggestion. we changed the sentence to:

"PM$_{2.5}$ is a long-term problem in some cities or regions." (Line 31)

3. Why Na was not measured? Could sea salt be a major source for PM$_{2.5}$ in Shenzhen, a coastal city?

**Response**: We did not measure Na mainly for two reasons.

First, it is difficult to accurately measure the online integration of Na, Na has lighter atomic number, so the uncertainty of data measured by X-ray fluorescence method is larger. The ACSM evaporator was ~550℃, and some components such as NaNO$_3$, NaCl, and Mg(NO$_3$)$_2$ in the sea salt may not be detected due to the lack of gasification at this temperature. Although the instruments of ion chromatography, such as MARGA can realize Na$^+$ online measurement, it is not suitable for integration due to its large volume and cumbersome operation.

Second, as the reviewer noted, sea salt represented by Na is not the main source of PM$_{2.5}$ in Shenzhen, previous studies have shown that sea salt contributes less (1% to 3%) to PM$_{2.5}$ (Huang et al., 2014; Huang et al., 2018).

We also pointed out it in the manuscript:

"The factor of sea salt cannot be identified in this study, because the measurement of its tracer (Na) is limited for X-ray fluorescence method and ACSM. However, the contribution of sea salt is little for Shenzhen (about 1% to 3%), and is not the main source of pollution." (Line 310-312)

4. 2.4 Design of the data analytics platform. More details for data conversion of each instrument should be given.

**Response**: Thanks for your comments as it identifies a potential area for confusion of other readers. The details of data analytics platform have been clarified in the revised supplement.

"Design of the data analytics platform

The design solution of the data analytics platform was shown in Fig. S2, the data of three online instruments (ACSM, Xact-625, AE-31) were processed in same (.csv) format and saved in their respective local computer using data transmission software, and uploaded data to a same SQL Server remote database for data management. The database was based on the dedicated server of the integrated system. The data of SHARP-5030i can be directly connected to the server. In addition, an atmospheric environment data collection and processing system had been established. The data in the SQL Server database was called to achieve unified processing and display of integrated data.

[Figure]

Figure S2. Design solution of data analytics platform"

5. Figure 2a. The mass closure is generally good. However, the authors should comment on some periods when a significant mass discrepancy appeared.

**Response**: As pointed out by the reviewer, the overall quality of the instrument is very good, with an average deviation of 6%, but the differences between reconstructed and measured are relatively large sometimes.

We further analyzed the 10/3 and 10/20, and think it may be related to the temperature. The temperature on October 3 is the highest during the observation period, and at

October 20, the temperature is at a relatively low level. The temperature has different effects on SHARP-5030i and integrated instruments, which results in different fitting effects to a certain extent. Of course, there may be other reasons, such as the measurement error of the integrated instruments, particle composition, temperature and relative humidity (Su et al., 2018; Zhang et al., 2017).

We also pointed out the reasons caused the underestimation in the manuscript:

"The average error margin of mass closure during the observation period is about 6%, which might be due to the measurement error of the integrated instruments, particle composition, temperature and relative humidity (Su et al., 2018; Zhang et al., 2017). A significant mass discrepancy between reconstructed and measured PM$_{2.5}$ appeared in some periods (Fig. 2a). For example, the underestimation on October 3 and the overestimation on October 20 occurred when the temperature was the highest and the lowest during the observation period, respectively. Therefore, it was speculated that temperature might affect the composition of PM$_{2.5}$, causing the mass closure to deviate." (Line 227-232)

6. Figure 4. The elements with low concentrations should be displayed with enlargement.

**Response**: We thank the reviewer for this suggestion. The figure is corrected as:

[Figure]

Figure 4. Concentrations of major chemical compositions of PM$_{2.5}$ measured by online Integrated system and offline sampling. Note that the EC in the online integrated system

was referred to as BC. (Line 273)

7. Figure 7. There is a spike at 1:00 am for both biomass burning and fugitive dust. Why?

**Response**: The spike of fugitive dust at 1:00 might be due to the influence of local sources. We carefully analyzed the time series of fugitive dust and biomass combustion sources, and found that the fugitive dust appeared a very high value at 1:00 on October 19, 25 and 28, which was significantly higher than the average dust concentration during the observation period, and made the average value of fugitive dust higher at 1:00. The tracer Ca also showed a higher value at the corresponding time, indicating that there were short-term dust emission activities. The biomass burning also appeared abnormally high value at 1:00 on October 2, relating to the firework shows on the China's National Day.

We added more specifically state the sources in manuscript:

"Analyzed with the time series of pollution sources (Fig. 5b), the results showed, the peaks around 22:00 and 1:00 of biomass burning was mainly due to the abnormally high source contribution, on the night of October 1, which is China's National Day, and there was a firework show 10 km away from the sampling site. Previous study has shown that the concentration of species (e.g. K, Ca, Cl$^-$, NO$_3^-$) in the PM$_{2.5}$ would have greatly increase due to the fireworks (Tsai et al., 2012). The spike at 1:00 of fugitive dust and biomass burning mainly caused by several abnormally high values at 1:00, suggesting it might be due to the influence of local short-term activities." (Line 355-361)

We added the standard deviation to the Fig. 7, so that readers can get more information from it.

[Figure]

Figure 7. Diurnal variations of the nine sources resolved by ME-2. (Line 368)

8. Conclusions. For a technical paper, the prospects of further development of the new system should be given.

**Response**: We have mentioned the prospects, and added more explanation about this in the manuscript:

"The development and successful application of the online integrated system and source apportionment method suggested that they can be used for precise regulation of PM$_{2.5}$. (such as fireworks)" (Line 416-417)

"That is, the high time resolution source analysis of the new integrated system is helpful to study the variation of the primary and secondary sources of PM$_{2.5}$ in the process of

heavy pollution, and to identify the key sources of heavy pollution and its mechanism, then to assess the impact of different sources. The system can run stably for a long time, and provides a key scientific basis for particulate matter control of China." (Lines 417-422)

**Reference**

Huang, X. F., Hui, Y., Gong, Z. H., Xiang, L., He, L. Y., Zhang, Y. H., and Min, H.: Source apportionment and secondary organic aerosol estimation of $PM_{2.5}$ in an urban atmosphere in China, Sci. China Earth Sci., 57, 1352–1362, https://doi.org/10.1007/s11430-013-4686-2, 2014.

Huang, X. F., Zou, B. B., He, L. Y., Hu, M., Prévôt, A.S.H., and Zhang, Y. H.: Exploration of PM2:5 sources on the regional scale in the Pearl River Delta based on ME-2 modeling. Atmos. Chem. Phys., 18(16), 11563-11580, http://doi.org/10.5194/acp-18-11563-2018, 2018.

Su, Y. S., Sofowote, U., Debosz, J., White, L., and Munoz, A.: Multi-Year Continuous $PM_{2.5}$ Measurements with the Federal Equivalent Method SHARP 5030 and Comparisons to Filter-Based and TEOM Measurements in Ontario, Canada, Atmosphere, 9(5), 191, https://doi.org/10.3390/atmos9050191, 2018.

Zhang, Y. J., Tang, L.L., Croteau, P. L., Favez, O., Sun, Y., Canagaratna, M. R., Wang, Z., Couvidat, F., Albinet, A., Zhang, H. L., Sciare, J., Prévôt, A.S.H., Jayne, J.T., and Worsnop, D. R.: Field characterization of the $PM_{2.5}$ Aerosol Chemical Speciation Monitor: insights into the composition, sources, and processes of fine particles in eastern China., Atmos. Chem. Phys., 17, 14501-14517, https://doi.org/10.5194/acp-17-14501-2017, 2017.

Tsai, H. H., Chien, L. H., Yuan, C. S., Lin, Y. C., Jen, Y. H., Ie, I. R.: Influences of fireworks on chemical characteristics of atmospheric fine and coarse particles during Taiwan's Lantern Festival, Atmospheric Environment., 62(2012), 256-264, http://doi.org/10.1016/j.atmosenv.2012.08.012, 2012.